# Focal Accumulation of ROS Can Block *Pyricularia oryzae* Effector BAS4-Expression and Prevent Infection in Rice

**DOI:** 10.3390/ijms21176196

**Published:** 2020-08-27

**Authors:** Yafei Chen, Sarmina Dangol, Juan Wang, Nam-Soo Jwa

**Affiliations:** 1Division of Integrative Bioscience and Biotechnology, College of Life Sciences, Sejong University, Seoul 05006, Korea; hnpp2010@163.com (Y.C.); sarminadangol@gmail.com (S.D.); gloria0826@163.com (J.W.); 2Department of Plant Physiology, Swammerdam Institute for Life Sciences, University of Amsterdam, 1090 GE Amsterdam, The Netherlands

**Keywords:** rice, immunity, *Pyricularia oryzae* (syn. *Magnaporthe oryzae*), ROS, effector

## Abstract

The reactive oxygen species (ROS) burst is the most common plant immunity mechanism to prevent pathogen infection, although the exact role of ROS in plant immunity has not been fully elucidated. We investigated the expression and translocation of *Oryza sativa* respiratory burst oxidase homologue B (OsRBOHB) during compatible and incompatible interactions between rice epidermal cells and the pathogenic fungus *Pyricularia oryzae* (syn. *Magnaporthe oryzae*). We characterized the functional role of ROS focal accumulation around invading hyphae during *P. oryzae* infection process using the OsRBOHB inhibitor diphenyleneiodonium (DPI) and the actin filament polymerization inhibitor cytochalasin (Cyt) A. OsRBOHB was strongly induced during incompatible rice–*P. oryzae* interactions, and newly synthesized OsRBOHB was focally distributed at infection sites. High concentrations of ROS focally accumulated at the infection sites and suppressed effector biotrophy-associated secreted (BAS) proteins BAS4 expression and invasive hyphal growth. DPI and Cyt A abolished ROS focal accumulation and restored *P. oryzae* effector BAS4 expression. These results suggest that ROS focal accumulation is able to function as an effective immune mechanism that blocks some effectors including BAS4-expression during *P. oryzae* infection. Disruption of ROS focal accumulation around invading hyphae enables successful *P. oryzae* colonization of rice cells and disease development.

## 1. Introduction

Plant pathogen effectors facilitate parasitism by suppressing pathogen-associated molecular pattern (PAMP) perception or modifying plant physiology to support pathogen growth and colonization [1,2,3]. While, plant resistance is determined by immune receptors that recognize appropriate ligands to activate defense [4], such as PAMP-triggered immunity (PTI) and effector-triggered immunity (ETI) [5,6,7,8]. These two types of plant immune responses suppress pathogen infection and ultimately protect the plant through hypersensitive response (HR)-mediated cell death [9]. ROS burst which can be triggered by PTI and ETI [10], is related to the establishment of disease resistance [11]. The functions of reactive oxygen species (ROS) in plant defense include acting as a toxic chemical against pathogens, enhancing host physical resistance by strengthening cell walls [12,13], mediating defense gene activation, and activating HR cell death [14,15]. However, further work is necessary to completely define the additional roles of ROS in plant defense mechanisms and development processes.

Plant respiratory burst oxidase homologue (RBOHs) are involved in ROS generation and innate immunity [16]. It has been reported that at least 11 genes encoded *Oryza sativa* respiratory burst oxidase homologue (OsRBOH) proteins in the rice genome, and *OsRBOH* genes were differentially expressed depending on the rice tissue and environmental stresses [17,18,19]. The localizations and enzyme activities of OsRBOH proteins affect ROS distribution during defense responses. The RBOHs inhibitor DPI significantly inhibited ROS generation in rice cells under biotic and abiotic stresses [19,20]. It has been proved that reduced nicotinamide adenine dinucleotide phosphate (NADPH) oxidase (NOX) complex associates/co-localizes with actin in animal cells, and the actin cytoskeleton has an important role in facilitating NOX enzyme activation at distinct cellular locations to induce focused ROS generation [21,22,23].

Plant pathogens have developed numerous direct and indirect effectors that suppress ROS production to maintain pathogenicity through long time coevolution [24]. Several fungal pathogens secrete extracellular catalase on the mycelial surface during host invasion, which detoxifies ROS produced by host cells [15,25]. Analysis of plant cell membranes during mycelial invasion proved that invasive hyphae penetrating the cell were sealed within extra-invasive hyphal membrane (EIHM) derived from host cell, and were not in direct contact with the intracellular matrix [26]. Previous studies showed that biotrophy-associated secreted (BAS) proteins were highly expressed during compatible interactions, but suppressed during incompatible interactions. BAS1 and BAS2 proteins accumulated in biotrophic interfacial complexes (BIC), BAS3 localized in BICs and penetration sites, and BAS4 distributed evenly along the invasive hyphae [27,28,29]. However, fungal effector secretion and translocation to plant cells remains largely unknown due to lack of functional studies. And more information should be provided to explain plant defense mechanisms on suppressing pathogen virulence.

The interaction between rice and *Pyricularia oryzae (P. oryzae)* is a useful model system to examine the pathogenicity and immune responses between fungal pathogens and host plants. Incompatible rice plants produce apoplastic ROS bursts as the immediate defense response against *P. oryzae* infection [20,30]. Sustained ROS production can effectively inhibit *P. oryzae* infection through HR-mediated cell death [14]. However, infection of rice cells with virulent *P. oryzae* does not induce ROS accumulation at the plasma membrane or around invasive hyphae [20]. Recent study found that an iron- and ROS-dependent ferroptotic cell death occurs during an incompatible rice–*P. oryzae* interaction, which shared in plants and animals is strong enough to halt or kill pathogens [20,31].

This study examined the spatiotemporal production of ROS during incompatible interactions. We report that ROS focally accumulated around invasive *P. oryzae* hyphae in rice cells during incompatible rice–*P. oryzae* interactions. *OsRBOHB* expression levels strongly increased and OsRBOHB protein localized around invasive hyphae only during incompatible interactions. Distinct localization of OsRBOHB around the infection site was correlated to ROS accumulation around invasive hyphae and inhibited effector BAS4 expression during incompatible interactions. ROS focal accumulation completely disappeared in the presence of the specific RBOH inhibitor DPI. Cytochalasin A also inhibited ROS accumulation around incompatible *P. oryzae* invasive hyphae. By contrast, ROS focal accumulation around invasive hyphae did not occur during compatible interactions. These combined results suggest that altered localization of OsRBOHB at the infection sites enables high concentration of ROS to accumulate around the avirulent *P. oryzae* invasive hyphae during incompatible interactions, which can prevent the expression and secretion of pathogen effector BAS4 and promote rice immunity.

## 2. Results

### 2.1. Avirulent P. oryzae Infection Induced OsRBOHB Expression and Altered OsRBOHB Localization in Rice Cells

NADPH oxidases are major ROS producers in plants under normal and stress conditions [32,33]. OsRBOHB has apoplastic ROS production activity in rice cells [34]. We compared *OsRBOHB* expression and protein localization patterns during *P. oryzae* infection in compatible and incompatible interactions. *OsRBOHB* expression in incompatible rice cv. HY sheath epidermal cells increased gradually after inoculation with avirulent *P. oryzae* INA168, and reached the highest level at 18 h post inoculation (hpi) (Figure 1A). Even though it started to decrease after 18 hpi, the expression level was still significantly higher than the normal condition. By contrast, inoculation of virulent *P. oryzae* PO6-6 in HY sheath epidermal cells resulted in continuous lower *OsRBOHB* expression levels than that under normal conditions (Figure 1A).

OsRBOHB is distributed in the plasma membrane of rice cells [35] and involves in ROS burst by receiving NADPH from the intracellular OsNADP-malic enzyme during pathogen infection [24]. OsRBOHB:GFP was delivered into rice cells to examine the distribution of OsRBOHB after infection with *P. oryzae*. OsRBOHB was evenly distributed in plasma membrane of HY sheath epidermal cells after mock treatment and inoculation with virulent *P. oryzae* PO6-6 (Figure 1B). By contrast, the invasion of avirulent *P. oryzae* INA168 in HY epidermal cells induced the specific localization of OsRBOHB:GFP around the invasive hyphae (Figure 1B), specifically localized in the front part of the invasive hyphae (primary hyphae), not in entire invasive hyphae. This change in protein localization may be a type of defense response in which OsRBOHB only accumulates focally around invasive mycelium during avirulent *P. oryzae* INA168 interactions. This unique distribution pattern of OsRBOHB ensures high concentration ROS accumulation at the site of invading hyphae rather than diffusion throughout the cell.

### 2.2. Avirulent P. oryzae Infection Induced ROS Focal Accumulation

Penetration of virulent *P. oryzae* hyphae into rice tissue invaginates the plasma membrane, and the invading hyphae expand within the first occupied epidermal rice cell, thereby perturbing plant hormone signaling before invading surrounding cells [36,37]. To analyze ROS accumulation, ROS quantity and distribution were detected in rice cells during compatible and incompatible interactions with *P. oryzae*. This study used CM-H_2_DCFDA staining to determine the ROS distribution in living cells without damaging the cell structure (which is a potential problem with 3,3′-diaminobenzidine (DAB) staining). The green fluorescence was produced by ROS reaction with CM-H_2_DCFDA in plant cells, but not the autofluorescence of plant phenolic compounds or cell-wall appositions around the invasive hyphae (Appendix A). And it was concentrated around the primary invasive hyphae in HY sheath cells infected with avirulent *P. oryzae* INA168, whereas the DAB method uniformly stained ROS throughout infected cells (Figure 2A). Higher levels of ROS were produced in HY cells during incompatible interactions with *P. oryzae* INA168 than by compatible interactions with *P. oryzae* PO6-6 (Figure 2B). Invasive mycelial growth in rice cells infected with avirulent *P. oryzae* was inhibited under high ROS conditions. Avirulent *P. oryzae* invasive hyphae displayed growth inhibition, appearing thin and filamentous and without bulbous branches (Figure 2A and Appendix A). By contrast, no ROS was detected by DAB staining or CM-H_2_DCFDA staining in compatible rice cells, and invasive hyphae grew actively, developed strong bulbous hyphae, and successfully colonized infected cells and neighbor cells (Figure 2A and Appendix A). These results indicate that ROS focal accumulation around invasive hyphae has an essential role in rice immunity through growth inhibition of *P. oryzae* invasive hyphae.

### 2.3. ROS Focal Accumulation around Invasive Hyphae Suppressed Effector BAS4 Expression

*P. oryzae* hyphae secrete effectors that suppress plant defense responses for successful colonization in rice cells [30,38,39]. If ROS focal accumulation inhibits *P. oryzae* hyphal growth and inhibits effector expression, it would be a much more effective defense mechanism. To investigate effector expression during *P. oryzae* infection process, the effector BAS4:eGFP was constructed and transformed into *P. oryzae* INA168. The BAS4:eGFP effector was normally expressed in *P. oryzae* invasive hyphae in compatible interaction with NB sheath epidermal cells (Figure 3A). By contrast, no fluorescence was detected in incompatible HY cells inoculated with avirulent *P. oryzae* INA168: BAS4:eGFP (Figure 3A).

The effect of ROS accumulation on BAS4:eGFP effector expression was observed using CellROX staining, which produces red fluorescence when it reacts with ROS in infected living cells (Figure 3A). In the compatible interaction, BAS4:eGFP expression occurred normally in the invasive hyphae of *P. oryzae*, but no BAS4:eGFP fluorescence was detected in the incompatible interaction (Figure 3A). Only red fluorescence from reaction with ROS was observed around the invasive hyphae of avirulent *P. oryzae* INA168:BAS4:eGFP, but not green fluorescence from BAS4:eGFP, indicating that ROS accumulation around invasive hyphae suppressed BAS4:eGFP expression (Figure 3A). ROS quantification data confirmed that ROS production in incompatible HY cells was much higher than in compatible NB cells after inoculation with *P. oryzae* INA168 (Figure 3B). These combined results indicated that ROS focal accumulation around avirulent *P. oryzae* invasive hyphae suppressed effector BAS4 expression.

### 2.4. DPI Suppressed ROS Production but Restored Effector BAS4 Expression

Plasma membrane–bound NADPH oxidase is a representative enzyme that rapidly generates ROS bursts and activates defense signaling during pathogen infection [18,19,32,33]. DPI suppresses ROS production by inhibiting NADPH oxidases [40,41,42]. DPI treatment significantly reduced ROS accumulation in avirulent *P. oryzae*-infected rice sheath epidermal cells, which confirmed that OsRBOHB-mediated ROS focal accumulation had a key role in rice defense responses (Figure 4). ROS focal accumulation around invasive hyphae occurred in incompatible rice–*P. oryzae* INA168 interaction at 36 hpi, but disappeared after DPI treatment (Figure 4A). As the ROS focal accumulation vanished, rice cells failed to induce normal HR responses to inhibit avirulent *P. oryzae* infection (Figure 4A,E). The invasive hyphae grew in normal pattern, developed multiple bulbous branches, restored effector BAS4:eGFP expression from avirulent *P. oryzae* INA168:BAS4:eGFP, and recovered pathogenicity (Figure 4B). DPI treatment also suppressed OsRBOHB expression in HY cells infected with avirulent *P. oryzae* INA168 (Figure 4C), which suppressed ROS production (Figure 4D) and disrupted ROS focal accumulation in incompatible rice cells (Figure 4A).

The effects of different DPI concentrations were observed to determine whether ROS focal accumulation is a key mechanism that suppresses effector BAS4:eGFP expression. As the DPI concentration increased from 0 to 5 μM, total ROS production and focal accumulation gradually decreased (Appendix A). At the same time, BAS4:eGFP expression restored from concentration-dependent growth of avirulent *P. oryzae* invasive hyphae (Appendix A), and the percentage of healthy invasive hyphae increased from 8% to 60% (Appendix A). These combined results suggest that ROS focal accumulation has a pivotal role in maintaining plant immunity by effectively inhibiting effector BAS4 expression in avirulent *P. oryzae* invasive hyphae.

### 2.5. Cytochalasin a Suppressed ROS Focal Accumulation and Restored Effector BAS4 Expression

Cytochalasin inhibits actin microfilament polymerization that is required for intracellular vesicle trafficking in plants, and Cytochalasin E (Cyt E) treatment suppresses iron-dependent ROS accumulation in rice sheaths during avirulent *P. oryzae* infection [20,43,44]. We investigated whether Cyt A affects ROS focal accumulation and HR cell death in HY sheath cells during avirulent *P. oryzae* INA168 infection. In incompatible interaction with *P. oryzae* INA168, ROS focally accumulated around the invasive hyphae and strong HR cell death occurred in HY cells. By contrast, ROS focal accumulation disappeared after Cyt A treatment (Figure 5A), and incompatible HY cells did not develop normal HR responses to avirulent *P. oryzae* infection (Figure 5A,D). At the same time, the growth of invasive hyphae and BAS4:eGFP effector expression in avirulent *P. oryzae* INA168:BAS4:eGFP was restored (Figure 5B), and the pathogen successfully colonized rice cells (Figure 5D). ROS quantification data showed that ROS production in HY cells slightly decreased after Cyt A treatment, but did not significantly differ from that in mock (Figure 5C). These combined results indicated that Cyt A treatment suppressed ROS focal accumulation around invasive hyphae and restored pathogen effector BAS4 expression to establish compatibility against avirulent *P. oryzae* infection.

## 3. Discussion

Fungal pathogen invasion into plant cells causes plant-derived extrahaustorial membrane (EHM) to surround the haustoria [45,46] or induces ROS focal accumulation around the invasive hyphae in incompatible rice–*P. oryzae* interactions [20]. In this study, we demonstrated that a majority of OsRBOHB was localized around the invasive hyphae in incompatible rice–*P. oryzae* interactions, but this OsRBOHB re-distribution did not occur in compatible rice–*P. oryzae* interactions (Figure 1B). As a result, distinct ROS focal accumulation occurred during the incompatible interaction but did not occur during the compatible interaction (Figure 2A). Based on our findings, we propose that ROS focal accumulation around invasive hyphae can have an effective role in inhibiting pathogen infection.

Plants express several RBOHs, which are all involved in ROS production although their cellular distribution and expression vary under different conditions [17,33,47,48,49]. OsRBOHB localizes in the plasma membrane and functions in the ROS burst during pathogen invasion [50]. In the present study, *OsRBOHB* expression increased only during the incompatible interaction, indicating that virulent pathogen inhibited *OsRBOHB* expression (Figure 1A). *OsRBOHB* was strongly expressed in incompatible rice cells, which ensures abundant ROS accumulation to defend against infection by avirulent pathogens. Our previous study demonstrated that ROS accumulated more strongly around invasive hyphae during the avirulent *P. oryzae* invasion of rice cells [20]. However, little is known about the microscopic changes of OsRBOHB distribution that occur during pathogen invasion. We showed that OsRBOHB was evenly distributed throughout the plasma membrane during pathogen-free or compatible rice–*P. oryzae* interactions (Figure 1B). By contrast, high OsRBOHB protein levels strongly accumulated around invasive hyphae and partially localized to the plasma membrane during incompatible rice–*P. oryzae* interactions (Figure 1B). Later, ROS distribution matched with the localization of OsRBOHB:eGFP protein around the primary invasive hyphae. It indicated that the OsRBOHBs around the penetration site were functional and effective in inhibiting *P. oryzae* hyphal growth and development by facilitating focal accumulation of high ROS.

ROS burst is one of the earliest host defense responses to pathogen infections [51]. ROS production in infected cells can be measured using different detection methods. The commonly used DAB staining method [52,53] destroys cell organelles, making it difficult to preserve the actual ROS accumulation pattern in living cells [20], but showed evenly distributed ROS within the dead cell (Figure 2A). We found that incompatible rice–*P. oryzae* interactions produced much higher ROS levels than compatible interactions (Figure 2B). CM-H_2_DCFDA staining in living rice cells revealed that a large amount of ROS uniquely accumulated around invasive hyphae only during incompatible interactions (Figure 2A). During incompatible rice–*P. oryzae* interactions, ROS accumulation around invasive hyphae appears from 12 h after inoculation and continues for 36 h, disappearing when HR cell death occurs [20]. Inhibition of OsRBOHB enzyme activity with DPI treatment significantly reduced ROS accumulation, with the ROS focal accumulation disappearing around invasive hyphae (Figure 4). DPI negatively affects plant development and defense responses by suppressing RBOH-mediated ROS generation. Dosage-dependent DPI treatment during incompatible interactions gradually reduced the total ROS level and focal accumulation (Appendix A). In the interaction between Lettuce cells and *Pseudomonas syringae* pv *phaseolicola*, highly localized accumulation of ROS caused localized membrane damage at sites of bacterial attachment, and DPI treatment reduced ROS quantity in lettuce cells [54].

The zig-zag hypothesis proposes that plant pathogens have co-evolved with plants by constantly developing various effectors to overcome host immunity for survival [9]. Plant pathogens effectively inhibit host immune targets involved in various pathways by releasing effectors into the plant cytoplasm during invasion [1,16]. BAS4 protein has been found to be expressed and distributed evenly along the invasive hyphae [27]. Although inhibition of a single BAS4 expression cannot represent the same inhibition of numerous effector secretions from *P. oryzae*, it might provide a possible ROS-mediated immune mechanism through effector suppression.

We found that ROS focally accumulated around invasive hyphae during incompatible rice–*P. oryzae* interactions and suppressed effector BAS4:eGFP expression (Figure 3, Figure 4 and Figure 5). In a previous study, the invasive hyphae of avirulent *P. oryzae* appeared thin and elongated with almost no branches until HR cell death occurred. By contrast, virulent *P. oryzae* actively colonize rice cells with bulbous hyphae that have multiple branches and invade surrounding cells during successful infections [26,27,28]. In the present study, the invasive hyphae of avirulent *P. oryzae* appeared thin and filamentous (Figure 2 and Figure 3, Appendix A), consistent with the previous study. Dosage-dependent DPI treatments during incompatible interactions recovered *P. oryzae* hyphal growth and BAS4:eGFP effector expression (Appendix A). These results suggest that ROS focal accumulation by OsRBOHB can inhibit *P. oryzae* invasion and suppress effector BAS4 expression. Furthermore, most of the protein biosynthesis including BAS4 might be halted under strong focal ROS accumulation condition resulting in suppression of IH growth and disappearance of BAS4:eGFP in incompatible interaction. However, the occurrence of HR cell death under ROS focal accumulation means that effectors expression is not completely inhibited, but rather suppressed below the level of disease development. The avirulent *P. oryzae* induced HR did not occur in rice cells treated with DPI (Figure 4), probably because pathogen effectors were expressed normally and were transferred into invading cells in the ROS-free environment.

Actin polymerization contributes to the modification of calcium channel activity and NADPH oxidase function [55,56,57]. Plant NADPH oxidases are localized at the plasma membrane and endometrium via vesicle trafficking along actin microfilaments under normal conditions and salinity stress [24,58,59]. Plant pathogen infection causes dynamic actin microfilament rearrangements at the site of cellular invasion [60,61,62,63]. Cytochalasin interferes with actin microfilament polymerization by inhibiting the rate of elongation, and acts on cytoplasmic streaming in plant cells [64,65,66,67,68,69,70] and suppresses HR cell death by avirulent *P. oryzae* [20]. It has high affinity binding to the barbed of F-actin to prevent the monomer addition [65]. It also inhibited F-actin assembly in bundles and meshwork, and NOX2 subunit localization was disrupted in the region of actin clearance [71]. In the present study, the actin polymerization inhibitor Cyt A [72,73] helped to establish the compatibility and colonization of avirulent *P. oryzae* in rice cells (Figure 5A,B). Cyt A abolished ROS focal accumulation and restored the expression of pathogen effector BAS4. However, the ROS quantity in infected rice cells decreased only slightly after Cyt A treatment (Figure 5C). This suggests that Cyt A does not directly inhibit the enzyme activity of OsRBOHB already present in the plasma membrane, but rather interrupt the distribution of newly synthesized OsRBOHB due to actin microfilament reorganization [43], so that the immune response cannot effectively occur at infection sites. Activity levels of Cytochalasins can be a little bit different depending on diverse biological properties even though their activities are basically similar. Cyt A also has antimicrobial activity that inhibits *Bacillus subtilis* and *Botrytis cinerea* growth and blocks ferric ion accumulation around the infection sites in wheat [74]. In this paper, when rice cultivar HY was inoculated with *P. oryzae* avirulent race INA168, Cyt A showed more stable suppression on HR cell death.

Based on the data presented in this study, we propose the hypothesis of ROS focal accumulation–mediated suppression of *P. oryzae* effector BAS4 expression during incompatible rice–*P. oryzae* interactions (Figure 6). OsRBOHB proteins are evenly distributed throughout the rice plasma membrane under pathogen-free normal conditions, but abundant OsRBOHB proteins are localized surrounding invasive hyphae during avirulent pathogen infection (Figure 1B). This re-distribution of OsRBOHB enables the formation of strong ROS focal accumulation within a very limited space at the site of *P. oryzae* infection. ROS focal accumulation can inhibit invasive hyphal growth and *P. oryzae* effector BAS4 expression. By contrast, OsRBOHB localization is not altered at the infection site of compatible interactions (Figure 1B), and ROS focal accumulation is blocked, effectors were successfully expressed and secreted into rice cells to inhibits rice targets associated with ROS production and other plant immune mechanisms [24]. If DPI inhibits OsRBOHB activity or Cyt A inhibits OsRBOHB re-distribution in incompatible interactions, ROS focal accumulation disappears and pathogens successfully colonize rice cells (Figure 4 and Figure 5). These combined results suggest that the formation of ROS focal accumulation can be an effective plant immune mechanism to inhibit some effectors including BAS4 expression and secretion from invasive hyphae during the invasion of *P. oryzae*. We hypothesize that plants can protect themselves from pathogen infection by strategically accumulating ROS, thereby blocking pathogen effector secretion. These results will improve our understanding of how plants maintain immunity during the co-evolution of plants and pathogens.

## 4. Materials and Methods

### 4.1. Plant Materials and Growth Conditions

The seeds of rice (*Oryza sativa* L.) cultivars Nipponbare (NB) and Hwayoungbye (HY) were provided by the National Institute of Crop Science, Jeonju, Korea (http://www.nics.go.kr). Rice seeds were soaked in water at 28 °C under continuous light condition (80 μmol photons m^−2^ sec^−1^) for 4–5 days (d). The germinated seeds were planted into pots (diameter = 12 cm, height = 11 cm) containing Baroker soil for rice (Seoul Bio, Gangnam-gu, Seoul, Korea), and were grown in a growth chamber with the following settings: 28 °C, 60% relative humidity, white fluorescent light (150 µmol photons m^−2^ s^−1^), and 16 h/8 h day/night photoperiod.

### 4.2. Fungal Culture Conditions

Two strains of the rice blast fungus *P. oryzae* (INA168 and PO6-6) were obtained from the Center for Fungal Genetic Resources (Seoul National University, Seoul, Korea; http://genebank.snu.ac.kr). *P. oryzae* INA168 is avirulent and *P. oryzae* PO6-6 is virulent to rice cv. HY, whereas rice cv. NB is susceptible to *P. oryzae* INA168. All fungal strains were stored at −20 °C and cultured on rice bran agar medium [30]. To produce aerial mycelia and harvest fungal spores, *P. oryzae* INA168 was grown under continuous light conditions (80 µmol photons m^−2^ s^−1^) at 25 °C for 12–14 d, whereas *P. oryzae* PO6-6 was grown under dark conditions at 25 °C for 12–14 d. Aerial mycelia were collected and sporulation was induced by incubating under continuous light conditions (80 µmol photons m^−2^ s^−1^) for 3–4 d. Sterilized water with 0.025% (*v*/*v*) Tween 20 (Sigma-Aldrich, St. Louis, MO, USA) was added to plates to collect the new produced spores, and the spore suspension was adjusted to appropriate concentrations for different experiments.

### 4.3. Fungal Transformation

The pBV551 plasmid containing the BAS4:eGFP cassette was provided by the Center for Fungal Genetic Resources (Seoul National University, Seoul, Korea; http://genebank.snu.ac.kr). The BAS4:eGFP:pBV551 contains the *hygromycin B* phosphotransferase gene as a selection marker and the enhanced GFP (eGFP) gene for labeling transformants. The BAS4:eGFP cassette was transformed into *P. oryzae* INA168 by performing polyethylene glycol (PEG)-mediated protoplast transformation as described previously [30]. Briefly, *P. oryzae* INA168 mycelia were cultured in broth medium (1 g sucrose, 0.6 g yeast extract, and 0.6 g casamino acid in 100 mL) for 3 d, and then all mycelia were incubated with 5 mg mL^−1^ Lysing Enzymes (Sigma-Aldrich, St. Louis, MO, USA) for 3–4 h using a platform shaker under room temperature. Then, 10 µg plasmid BAS4:eGFP:pBV551 was transformed into freshly harvested *P. oryzae* mycelial protoplasts (1 × 10^7^ mL^−1^) as described previously [30]. Positive transformants were selected on TB3 agar medium supplemented with hygromycin B (20 g sucrose, 0.3 g yeast extract, 0.3 g casamino acid, 1 g glucose, 0.8 g agar, and 0.2 mg mL^−1^ hygromycin B in 100 mL), and confirmed by polymerase chain reaction (PCR) using BAS4:eGFP primers. The primers used in this study are listed in Appendix A. The positive fungal transformants were confirmed by visualizing eGFP fluorescence using a fluorescence microscope (Zeiss equipped with Axioplan 2; Campbell, CA, USA) with GFP filter (488 nm excitation/505–550 nm emission).

### 4.4. Plasmid Construction and Subcellular Localization

The *OsRBOHB* (LOC_Os01g25820) coding region was amplified from a rice cDNA library with gene-specific primers containing *Eco*R1 and *Xba*1 adapter sites. The primers used in this study are listed in Appendix A. The PCR product was ligated to pBlueScript II SK (pBSK) and transferred into the destination vector pCAMBIA1304 tagged with GFP. The final clone was confirmed by digestion and nucleotide sequencing (Macrogen, Seoul, Korea).

Subcellular localization of OsRBOHB in the rice epidermal layer was analyzed using gene gun method as described previously [20,30]. Briefly, 10 µg OsRBOHB:pCAMBIA1304 plasmid was bombarded into rice sheath epidermal cells placed in Murashige and Skoog medium (2.15 g Murashige and Skoog, 15 g sucrose, and 4 g Gellan gum in 1 l Milli-Q water) using a biolistic particle delivery system (PDS-1000/He™ System, Bio-Rad, Hercules, CA, USA), followed by incubation at 25 °C for 36 h in the dark. Images were captured using confocal laser microscopy (Leica, TCS SP5, Mannheim, Germany) with bright field and GFP filter (488 nm excitation/505–550 nm emission).

To detect OsRBOHB localization after pathogen infection, 10 µg OsRBOHB:pCAMBIA1304 plasmid was bombarded into rice sheath epidermal cells as described above, followed by incubation in the dark at 25 °C for 12 h. Freshly harvested *P. oryzae* INA168/PO6-6 spore suspensions were inoculated on the rice epidermal layer surface, and then incubated for 24 h in the dark at 25 °C. Images were captured using confocal laser microscopy (Leica, TCS SP5, Mannheim, Germany) with GFP filter (Ex/Em: 488 nm/505–550 nm in wave length).

### 4.5. Pathogenicity Test

The sheath inoculation method was performed to test for pathogenicity as described previously [20,30]. A freshly harvested spore suspension (5.0 × 10^5^ mL^−1^) of *P. oryzae* (INA168, INA168:BAS4:eGFP, and PO6-6) was inoculated onto 4–5 cm rice leaf sheaths (*n* = 4) from 5–6-week old rice cv. NB and HY. The inoculated rice leaf sheaths were incubated at 25 °C in the dark for 48 h in a moistened chamber. The infection phenotype was determined in isolated chlorophyll-free thin epidermal layer using fluorescence microscopy (Zeiss equipped with Axioplan 2; Campbell, CA, USA) with bright field. The infected cells were categorized into two phenotypes: cells with invasive hyphae (IH) and cells with dead hyphae/hypersensitive response (HR). The infection assay and quantification of infected cells were performed at least three times with similar results.

### 4.6. Cytochalasin A and Diphenyleneiodonium Treatment

To suppress the production and focal accumulation of ROS in *P. oryzae*-infected rice cells, Cyt A (Cayman Chemical Company, Ann Arbor, MI, USA) and DPI (Sigma-Aldrich, St. Louis, MO, USA) treatments were performed as described previously with slight modification [20]. Briefly, 41.8 µM of Cyt A or different concentrations (1, 3, 5 µM) of DPI were mixed with the freshly harvested spore suspension (5.0 × 10^5^ mL^−1^) of *P. oryzae* INA168. The Cyt A- or DPI-treated spores were inoculated into rice leaf sheaths (*n* = 4) along with untreated spores (Mock). The effects of Cyt A and DPI were observed at 48 hpi using fluorescence microscopy (Zeiss equipped with Axioplan 2; Campbell, CA, USA).

### 4.7. ROS Detection

ROS accumulation in *P. oryzae*-infected rice epidermal cells was detected using 5-(and-6)-chloromethyl-2′,7′-dichlorodihydrofluorescein diacetate, acetyl ester (CM-H_2_DCFDA, Molecular Probes Life Technologies, Eugene, OH, USA), CellROX deep red reagent (Invitrogen, Life Technologies Corporation, Eugene, OR, USA), and DAB staining [20,75]. Rice leaf sheaths (*n* = 4) were inoculated with different *P. oryzae* strains, and then incubated in the dark for 36 h at 25 °C. The isolated rice epidermal layer was first soaked in water for 5 min at 4 °C to reduce wound-induced ROS, followed by staining with 2 µM CM-H_2_DCFDA or 5 µM CellROX deep red reagent in the dark for 30 min at room temperature. The stained epidermis was washed three times with 1× PBS buffer for 5 min, and the cells were observed using fluorescence microscopy (Zeiss equipped with Axioplan 2; Campbell, CA, USA) with bright field and green fluorescence filter (488 nm excitation/505–550 nm emission) to detect CM-H_2_DCFDA stained cells. Cells stained with CellROX deep red reagent were observed using confocal laser microscopy (Leica, TCS SP5, Mannheim, Germany) with bright field and red fluorescence filter (644 nm excitation/665 nm emission).

DAB staining was performed as described previously [20]. Rice sheath epidermal cells inoculated with *P. oryzae* were isolated at 48 hpi, followed by DAB staining (1 mg mL^−1^, Sigma-Aldrich, St. Louis, MO, USA) for 8 h at room temperature and destaining with ethanol:acetic acid:glycerol (3:1:1, *v*/*v*/*v*). Images were captured using fluorescence microscopy (Zeiss equipped with Axioplan 2; Campbell, CA, USA).

### 4.8. ROS Quantification

ROS was quantified by performing a chemiluminescence assay [20]. Rice sheaths (*n* = 4) were inoculated with freshly harvested spore suspensions of *P. oryzae*. At 48 hpi, the inoculated rice epidermal layers were cut into small pieces (0.5 × 0.2 cm) and submerged in Milli-Q water for 5 min at 4 °C to reduce wound-induced ROS. Each epidermal sample (0.5 × 0.2 cm) was transferred into 96-well plates containing 100 mL luminol buffer [30 µL luminol (Bio-Rad, Hercules, CA, USA), 1 µL horseradish peroxidase (Jackson Immunoresearch, West Grove, PA, USA), and 69 µL Milli-Q water], incubated for 5 min in the dark, and ROS was detected by luminometer under dark condition.

### 4.9. Gene Expression Analysis

Leaf sheaths of rice cv. HY (*n* = 4) inoculated with different *P. oryzae* spore suspensions were sampled at different time points (0, 3, 6, 12, 18, 24, 36, and 48 hpi). RNA extraction was performed using TRIzol reagent (Invitrogen, Carlsbad, CA, USA) according to the manufacturer’s protocol. Then, 2 µg of RNA was used to synthesize first-strand cDNA in 20 µL total reaction mixture using cDNA synthesis kit (Invitrogen, Carlsbad, CA, USA) according to the manufacturer’s protocol. *OsRBOHB* expression was determined using 1 µg cDNA and Real-Time Quantitative Reverse Transcription Polymerase Chain Reaction (Real-time qRT-PCR, Stratagene, Mx3000p, Santa Clara, CA, USA). The qRT-PCR assay was performed using TOPreal qPCR 2 × PreMIX (SYBR Green with low ROX, Enzynomics, Daejon, South Korea) according to the manufacturer’s protocol. The *Ubiquitin* (LOC_Os06g46770) transcript level was used to normalize the *OsRbohB* transcript level. Relative gene expression level was determined by comparing with the mock sample at 0 hpi. The primers used in this study are listed in Appendix A.

### 4.10. Statistical Analysis

All experiments were repeated independently more than three times. Each biological replicate contained independent leaf sheath samples taken from different rice plants. The number of biological replicates (*n*) is given in the Figure legends. Statistical analyses were performed using Prism 7 software (GraphPad, La Jolla, CA, USA).

## Figures and Tables

**Figure 1 ijms-21-06196-f001:**
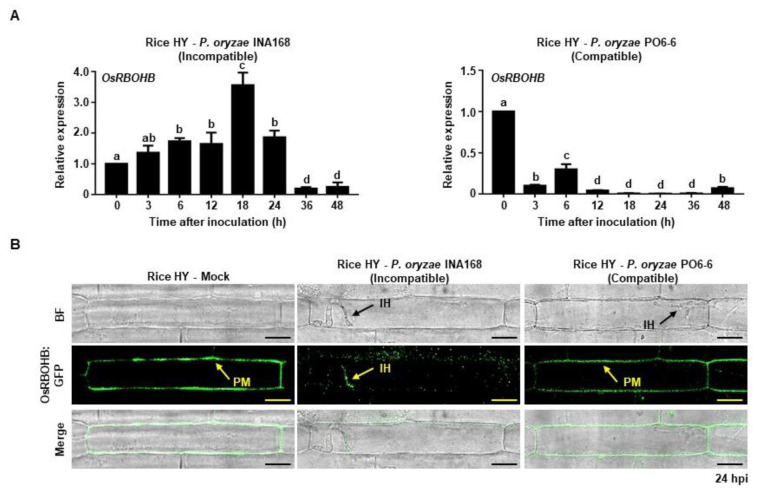
OsRBOHB- focally accumulated around invasive hyphae during incompatible rice–*P. oryzae* interaction. (**A**) Transcriptional analysis of *OsRBOHB* in rice leaf sheaths after inoculation with *P. oryzae* incompatible strain INA168 and compatible strain PO6-6. The *OsRBOHB* expression level was detected at different time points using quantitative real-time PCR (qRT-PCR) after normalizing with respect to the housekeeping gene *Ubiquitin*. The results were presented as means ± SD of relative expression levels from three independent experiments. Letters above the bars indicate significant differences in relative expression at different time points (one-way ANOVA, *p* < 0.05, *n* = 3). (**B**) Localization of GFP-tagged OsRBOHB in epidermal cells of rice cv. HY after inoculation with *P. oryzae* incompatible INA168 and compatible PO6-6. OsRBOHB:GFP fluorescence is indicated by arrows in rice cells. Scale bars = 10 µm. All images were captured using confocal laser microscopy with bright field or green fluorescence filter (488 nm excitation/505–550 nm emission). BF, bright field; IH, invasive hyphae; PM, plasma membrane.

**Figure 2 ijms-21-06196-f002:**
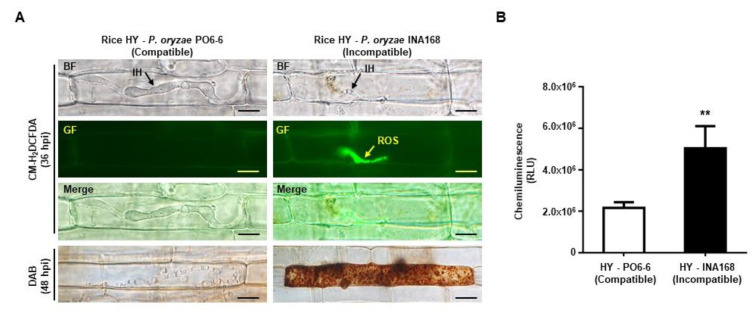
ROS accumulation in compatible and incompatible rice–*P. oryzae* interactions. (**A**) ROS distribution in HY cells during compatible and incompatible *P. oryzae* interactions. DAB and CM-H_2_DCFDA staining were performed to detect ROS distribution at 48 hpi and 36 hpi. (**B**) ROS quantification in HY sheath epidermal cells during compatible and incompatible rice–*P. oryzae* interactions. ROS was quantified by chemiluminescence assay. Values represent means ± SD of total relative luminescent units (RLU). All images were captured using fluorescence microscopy with bright field or green fluorescence filter (488 nm excitation/505–550 nm emission). BF, bright field; GF, green fluorescence; IH, invasive hyphae. Scale bars = 10 µm. Asterisks indicate significant differences between compatible and incompatible interactions (Student’s *t*-test, ** *p* < 0.01, *n* = 3).

**Figure 3 ijms-21-06196-f003:**
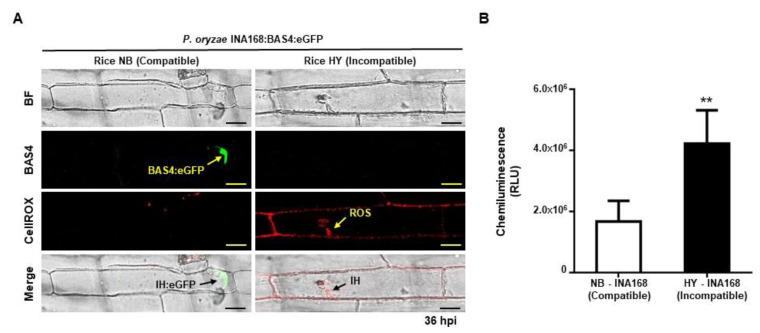
ROS accumulation inhibited the expression of BAS4 effector in *P. oryzae* INA168:BAS4:eGFP invasive hyphae in the incompatible interaction with rice cv. HY. (**A**) BAS4 expression and ROS accumulation in leaf sheaths of rice cv. NB and HY during *P. oryzae* INA168:BAS4:eGFP infection. CellROX staining was performed to detect ROS accumulation. Arrow indicates ROS accumulation and BAS4 secretion. All images were captured using confocal laser microscopy with bright field, green fluorescence filter (488 nm excitation/505–550 nm emission), or red fluorescence filter (644 nm excitation/665 nm emission). Scale bars = 10 µm. (**B**) Quantification of ROS in leaf sheaths of rice cv. NB and HY during *P. oryzae* INA168 infection. ROS was quantified by chemiluminescence assay. Values represent means ± SD of total relative luminescent units (RLU). Asterisks indicate significant differences among different rice sheaths (Student’s *t*-test, ** *p* < 0.01, *n* = 3). IH, invasive hyphae.

**Figure 4 ijms-21-06196-f004:**
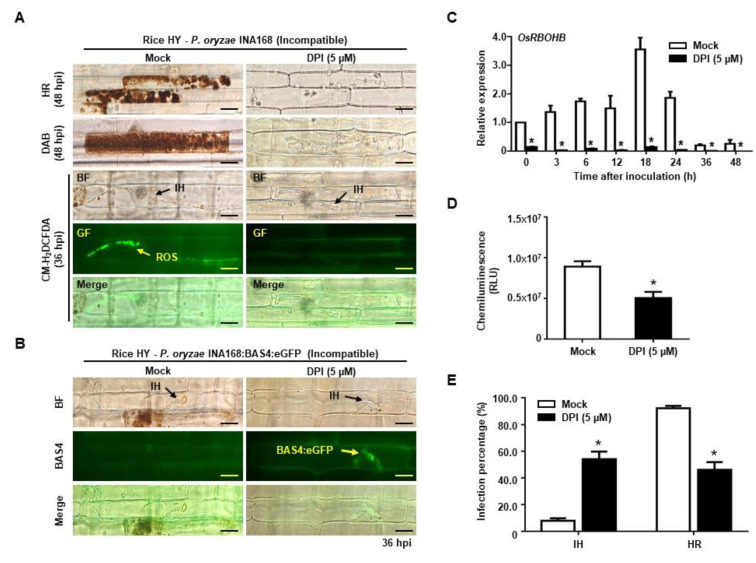
Inhibition of OsRBOHB by DPI treatment suppressed ROS accumulation and stimulated effector BAS4 expression during incompatible rice–*P. oryzae* interactions. (**A**) DPI treatment suppressed ROS accumulation in rice cv. HY cells inoculated with *P. oryzae* INA168. DAB and CM-H_2_DCFDA staining were performed to detect ROS accumulation in mock and DPI-treated rice cv. HY leaf sheaths. (**B**) Effector BAS4 expression of INA168:BAS4:eGFP invasion to DPI-treated leaf sheaths of rice cv. HY. (**C**) OsRBOHB expression in DPI-treated leaf sheaths of rice cv. HY inoculated with incompatible *P. oryzae* INA168. OsRBOHB expression at different time points was detected by qRT-PCR after normalizing with respect to the housekeeping gene *Ubiquitin*. Results are presented as means ± SD of the relative quantity from three independent repeats. (**D**) Quantification of ROS production in mock and DPI-treated leaf sheath cells of rice cv. HY after inoculation with incompatible *P. oryzae* INA168. ROS production was measured by chemiluminescence assay. Values represent means ± SD of total relative luminescent units (RLU). (**E**) Quantification of infection patterns in DPI-treated leaf sheath cells of rice cv. HY at 48 hpi with incompatible *P. oryzae* INA168. The percentages of two infection patterns are shown as mean values ± SD, *n* = 3 leaf sheaths from different plants. All images were captured using fluorescence microscopy with bright field or green fluorescence filter (488 nm excitation/505–550 nm emission). Scale bars = 10 µm. Asterisks indicate significant differences (Student’s *t*-test, * *p* < 0.01, *n* = 3). IH, invasive hyphae; HR, hypersensitive response; GF, green fluorescence; BF, bright field.

**Figure 5 ijms-21-06196-f005:**
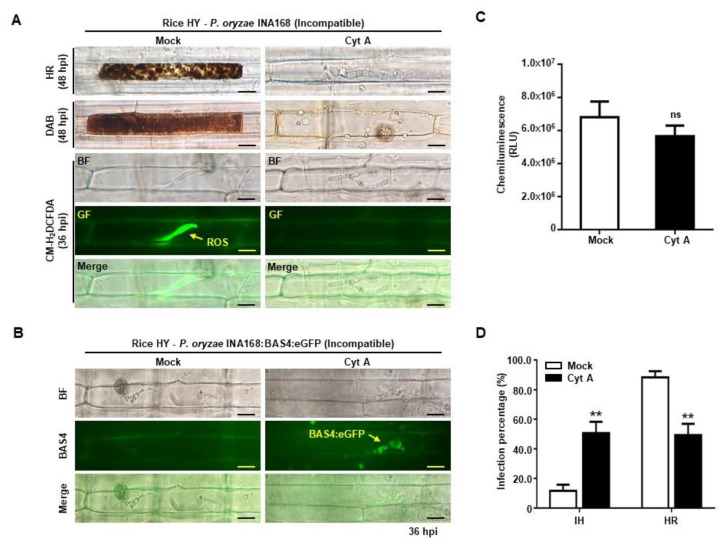
Cytochalasin A treatment suppressed ROS focal accumulation and restored BAS4 effector expression during incompatible rice–*P. oryzae* interactions. (**A**) Cyt A treatment suppressed ROS accumulation in rice cv. HY cells inoculated with *P. oryzae* INA168. ROS production and focal accumulation in mock and Cyt A-treated HY leaf sheaths cells were detected by staining with DAB and CM-H_2_DCFDA. (**B**) Successful expression of BAS4 effector in incompatible INA168:BAS4:eGFP invasive hyphae in Cyt A-treated leaf sheaths of rice cv. HY. (**C**) Quantification of ROS production in mock and Cyt A-treated leaf sheath cells of rice cv. HY after inoculation with incompatible *P. oryzae* INA168. ROS was quantified by chemiluminescence assay. Values represent means ± SD of total relative luminescent units (RLU). (**D**) Quantification of infection patterns in Cyt A-treated leaf sheath cells of rice cv. HY at 48 hpi with incompatible *P. oryzae* INA168. The percentages of two infection patterns are shown as mean values ± SD, *n* = 3 leaf sheaths from different plants. All images were captured using fluorescence microscopy with bright field or green fluorescence filter (488 nm excitation/505–550 nm emission). Scale bars = 10 µm. Asterisks indicate significant differences in infection patterns between mock and Cyt A treatment (Student’s *t*-test, ** *p* < 0.01, *n* = 3). IH, invasive hyphae; HR, hypersensitive response; GF, green fluorescence; BF, bright field. ns, no significance.

**Figure 6 ijms-21-06196-f006:**
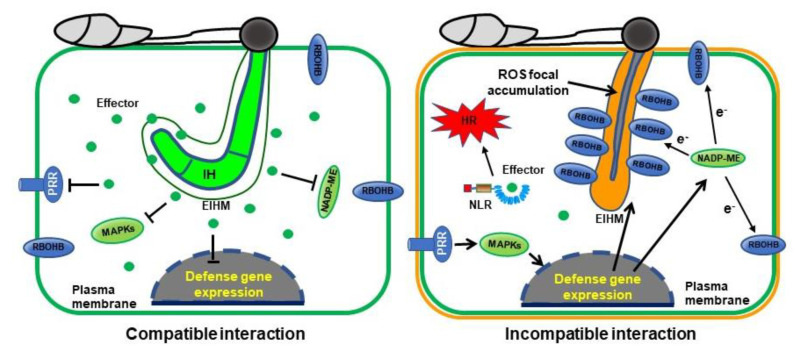
Model of ROS focal accumulation–mediated suppression on hyphal invasion and effector expression and secretion in rice–*P. oryzae* incompatible interactions. In compatible rice–*P. oryzae* interactions, invasive hyphae secrete effectors into the rice cytoplasm to interfere with PRR, MAPK cascades, and OsNADP-ME and OsRBOHB activities. Suppressed OsNADP-ME does not supply electron (e^−^) to OsRBOHB, which function in ROS production. However, OsRBOHB is strongly expressed, and many OsRBOHB localize around the invasive hyphae and produce considerable ROS (brown color) during avirulent *P. oryzae* infection. This ROS focal accumulation effectively inhibits invasive hyphal growth and some effector expression and secretion. Effector secretion is blocked in incompatible interactions, but the rarely leaked effectors are recognized by NLR and induce strong HR cell death to eliminate pathogens. PRR, pattern recognition receptors; NLR, nucleotide binding leucine-rich repeat; EIHM, extra invasive hyphal membrane; IH, invasive hyphae.

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
