# Peer review of "Focal Accumulation of ROS Can Block Pyricularia oryzae Effector BAS4-Expression and Prevent Infection in Rice"

_ijms, 2020, doi:10.3390/ijms21176196_

Round 1

Reviewer 1 Report

Authors investigated the expression and translocation of OsRBOHB during compatible and incompatible interactions between rice and blast fungus, and successfully showed that OsRBOHB was focally distributed at infection sites during incompatible interaction. In addition, they characterized the functional role of ROS focal accumulation using DPI and Cyt A. Focally accumulated ROS suppresses invasive hyphae growth and authors insist that it also suppresses an effector BAS4 secretion.

Although results about OsRBOHB and BAS4 are novel and informative, induction of ROS focal accumulation around the invasive hyphae in incompatible interactions has been reported in the previous paper (reference 20) by the same group. And results in Fig.2A, Fig.4A,D,E and Fig. 6A are similar to the previous work, although used rice cultivar, M. oryzae strain, type of cytochalasin are different.

Major comments

Title and throughout:

Authors describe ROS blocks or suppresses BAS4 secretion. However, their experiments do not discriminate BAS4 expression, production and secretion. In Fig.3, since BAS4:eGFP is not detected in incompatible interaction, I suppose the expression or production but not secretion of BAS4 would be blocked. If only secretion (into rice cells) is blocked, fluorescence in the fungi cells would be detected. Furthermore, I speculate that most of the protein biosynthesis (not limited to BAS4 or secreted proteins) in the invasive hyphae (IH) may be blocked in the incompatible interaction. For example, GFP protein in IH is not detected in the incompatible interaction in Fig.3A in reference 20, suggesting that most of the protein biosynthesis is suppressed or most of the proteins are degraded.

Authors should address these questions and revise the text.

Minor comments

Title and throughout: ‘Magnaporthe oryzae’ should be ‘Pyricularia oryzae’ or ‘Pyricularia oryzae (syn. Magnaporthe oryzae)’ according to the recommendation of International Commission on the Taxonomy of Fungi (c.f. IMA Fungus 7(1): 155–159 (2016)).

L20: cytochalasin (Cyt A) should be cytochalasin (Cyt) A

L91: add ‘rice cv.’ in front of HY.

L112: ‘with compatible M. oryzae strain PO6-6 and incompatible strain INA168’ should be ‘with incompatible P. oryzae strain INA168 and compatible strain PO6-6’

L117: ‘compatible M. oryzae PO6-6 and incompatible INA168’ should be ‘incompatible P. oryzae INA168 and compatible PO6-6’

L119 to 121: ‘Letters…… n=3)’ should be moved behind ‘independent experiments.’ in L115.

L136 &329: I understand thin, but it is difficult to understand ‘elongated, rarely underwent cell division’. If the authors insist so, further explanation is necessary.

L137: Figure 2B should be Figure 2A.

L139: ‘cell’ should be deleted.

L151 & 182: delete ‘*P<0.05, and’.

L179 to181: ‘All……10 um.’ should be moved behind ‘BAS4 secretion.’ in L177.

Figure 5: This figure should be move to Supplementary materials.

L321: change ‘can’ to ‘might’. At present, most of the protein biosynthesis (not limited to secreted proteins) in the invasive hyphae may be blocked.

L433: gen should be gene.

L501: accession number of the Ubiquitin should be added here or somewhere in the text.

Author Response

Comments and Suggestions for Authors

Major comments

Title and throughout:

Point 1. Authors describe ROS blocks or suppresses BAS4 secretion. However, their experiments do not discriminate BAS4 expression, production and secretion. In Fig.3, since BAS4:eGFP is not detected in incompatible interaction, I suppose the expression or production but not secretion of BAS4 would be blocked. If only secretion (into rice cells) is blocked, fluorescence in the fungi cells would be detected.

Response 1: We apologize for confusion of effector expression, production and secretion. Our data suggest that the expression and production of BAS4:eGFP in the invasive hyphae are significantly suppressed in incompatible interaction. Previous studies also proved that the fluorescence from BAS proteins was absent or severely attenuated in incompatible interaction (Mosquera et al., 2009). In this paper, we think the expression and secretion of BAS4:eGFP are tightly connected process, and ROS focal accumulation can be the main reason for the suppression of effector expression and following secretion. Reviewer 1’s comment about the detail role of ROS focal accumulation around P. oryzae invasive hyphae is more scientific and we applied his comment in the title and text. According to Reviewer1’s comment, we have changed “effector BAS4 secretion” to “effector BAS4 expression” in the title and in the Results and Discussion sections. (Title, and Line 24, 25, 164, 167-1171, 174, 176, 181, 185, 188-191, 202, 212, 220, 223-225, 240, 242, 246, 263, 273, 278, 282, 285-287, 363, 379, 390, 398, 403, 410, 416, 560-561 and 564 ).

Point 2. Furthermore, I speculate that most of the protein biosynthesis (not limited to BAS4 or secreted proteins) in the invasive hyphae (IH) may be blocked in the incompatible interaction. For example, GFP protein in IH is not detected in the incompatible interaction in Fig.3A in reference 20, suggesting that most of the protein biosynthesis is suppressed or most of the proteins are degraded.

Response 2: We agree with the Reviwer 1’s comment that most of the protein biosynthesis in the invasive hyphae (IH) may be blocked in the incompatible interaction. In this study, we used only BAS4:eGFP as an indicator of effector protein to explore the role of focal ROS accumulation in IH regarding effector expression and secretion. Our results clearly show that ROS focal accumulation significantly suppress IH growth, development and further colonization in incompatible interaction. As Reviewer1 indicated, most of the protein biosynthesis including BAS4 might be halted under strong focal ROS accumulation condition resulting in suppression of IH growth and disappearance of BAS4:eGFP in incompatible interaction. Thus, we have changed the text according to Revewer1’s comments (Line 358-362).

Minor comments

Point 3. Title and throughout: ‘Magnaporthe oryzae’ should be ‘Pyricularia oryzae’ or ‘Pyricularia oryzae (syn. Magnaporthe oryzae)’ according to the recommendation of International Commission on the Taxonomy of Fungi (c.f. IMA Fungus 7(1): 155–159 (2016)).

Response 3: Thank you very much for this updating. ‘Magnaporthe oryzae’ in Title, Figures and the whole text have been changed into Pyricularia oryzae (syn. Magnaporthe oryzae).

Point 4. L20: cytochalasin (Cyt A) should be cytochalasin (Cyt) A

Response 4: We are sorry for this mistake. “cytochalasin (Cyt A)” has been changed into cytochalasin (Cyt) A in Line 20.

Point 5. L91: add ‘rice cv.’ in front of HY.

Response 5: Thank you for your comment. ‘rice cv.’ has been added in front of HY in Line 94.

Point 6. L112: ‘with compatible M. oryzae strain PO6-6 and incompatible strain INA168’ should be ‘with incompatible P. oryzae strain INA168 and compatible strain PO6-6’.

Response 6: Thank you for this comment. ‘with compatible M. oryzae PO6-6 and incompatible INA168’ has been changed into ‘with P. oryzae incompatible strain INA168 and compatible strain PO6-6’ in text, and it fits with the figures after this modification (Line 116-117).

Point 7. L117: ‘compatible M. oryzae PO6-6 and incompatible INA168’ should be ‘incompatible P. oryzae INA168 and compatible PO6-6’

Response 7: Thank you for this comment, and it fits with the figures after this modification. ‘compatible M. oryzae PO6-6 and incompatible INA168’ has been changed into ‘P. oryzae incompatible INA168 and compatible PO6-6’ in text (Line 123-124).

Point 8. L119 to 121: ‘Letters…… n=3)’ should be moved behind ‘independent experiments.’ in L115.

Response 8: Thank you for this comment. The sentence “Letters above the bars indicate significant differences in relative expression at different time points (one-way ANOVA, P<0.05, n=3).” has been moved to Line 120-121.

Point 9. L136 &329: I understand thin, but it is difficult to understand ‘elongated, rarely underwent cell division’. If the authors insist so, further explanation is necessary.

Response 9: We are sorry for this confusing information in the text. Previous studies showed that the initial hyphae after penetration the cell wall was thin and filamentous (filamentous hyphae), and they enlarged into invasive hyphae (IH) in the invaded cells and developed bulbous IH which grown rapidly from one cell to neighbor cells (Kankanala et al., 2007). In our study, IH successfully develop bulbous IH and invade neighbor cells in compatible interaction, but in incompatible interaction, the initial thin filamentous hyphae hardly develop to bulbous IH, and limited in first invaded cell without any branches. To precisely describe the hyphae pattern in our manuscript, we have changed these two sentences into “Avirulent P. oryzae invasive hyphae displayed growth inhibition, appearing thin and filamentous and without bulbous branches (Line 145-146)”, and “the invasive hyphae of avirulent P. oryzae appeared thin and filamentous (Figures 2,3,S1), consistent with the previous study (Line 355-356) ”.

Point 10. L137: Figure 2B should be Figure 2A.

Response 10: We apologize for this mistake in text. “Figure 2B” in Line 146 has been changed to “Figure 2A”. Thank a lot.

Point 11. L139: ‘cell’ should be deleted.

Response 11: We are sorry for this mistake in text. “cell” in Line 149 has been deleted.

Point 12. L151 & 182: delete ‘*P<0.05, and’.

Response 12: Thank you for this comment. In text, “*P<0.05, and” in Line 163 and Line 201 have been deleted.

Point 13. L179 to181: ‘All……10 um.’ should be moved behind ‘BAS4 secretion.’ in L177.

Response 13: Thank you for this comment. The sentence “All images were captured using confocal laser microscopy with bright field, green fluorescence filter (488 nm excitation/505–550 nm emission), or red fluorescence filter (644 nm excitation /665 nm emission). Scale bars=10 µm.” has been moved to Line 193-195 in text.

Point 14. Figure 5: This figure should be move to Supplementary materials.

Response 14: Thank you for this comment. In text, Figure 5 has been transferred to Supplementary material as “Figure S3” (Line 559-574), and the label of “Figure 6 and Figure 7” has been changed to “Figure 5 and Figure 6”. All the numbers of these figures has been changed correspondingly in the whole text.

Point 15. L321: change ‘can’ to ‘might’. At present, most of the protein biosynthesis (not limited to secreted proteins) in the invasive hyphae may be blocked.

Response 15: We really appreciate for your correction and agree with your comment. In text, “can” has been changed into “might” in Line 347.

Point 16. L433: gen should be gene.

Response 16: We are sorry for this spell mistake. “gen” has been changed into “gene” in the text, Line 467.

Point 17. L501: accession number of the Ubiquitin should be added here or somewhere in the text.

Response 17: We apologize for the missing accession number of the Ubiquitin. “Ubiquitin (LOC_Os06g46770) has been added in materials and methods section in Line 535-536.

Reviewer 2 Report

Manuscript deals with the interaction between rice and the pathogen Magnaporthe oryzae. Particularly, the work paid attention to the localisation of an enzyme involved in pathogen-associated oxidative burst and, through the use of selective inhibitors, provide evidences that ROS burst focused around the invasive iphae is required during incompatible interaction. ROS burst activated by plant also inhibits the production by pathogen of secreted protein (biotrophy-associated secreted) that play a role in tissue colonization.
The article is interesting and science is sounding, experimental set is correct, data are well presented and supported by images, discussion is clear.
In my opinion there are no lacks in experimentation and analysis, therefore it is worthwhile to be accepted in present form, however, minor spell mistakes should be fixed (i.e. line 409; Fungal Transfermation)

Author Response

Comments and Suggestions for Authors

Manuscript deals with the interaction between rice and the pathogen Magnaporthe oryzae. Particularly, the work paid attention to the localisation of an enzyme involved in pathogen-associated oxidative burst and, through the use of selective inhibitors, provide evidences that ROS burst focused around the invasive hyphae is required during incompatible interaction. ROS burst activated by plant also inhibits the production by pathogen of secreted protein (biotrophy-associated secreted) that play a role in tissue colonization. The article is interesting and science is sounding, experimental set is correct, data are well presented and supported by images, discussion is clear.

Point 1. In my opinion there are no lacks in experimentation and analysis, therefore it is worthwhile to be accepted in present form, however, minor spell mistakes should be fixed (i.e. line 409; Fungal Transfermation)

Response 1: We are sorry for this spell mistake. “Fungal Transfermation” has been changed into “Fungus Transformation” in the text, Line 443. Thanks a lot for review’s comment.

Round 2

Reviewer 1 Report

The revised paper properly responded to my comments. Therefore, I do not have additional comments.